# SAMPLE IMPORTANCE IN SGD TRAINING

## ABSTRACT

Deep learning requires increasingly bigger models and datasets to improve generalization on unseen data, where some training data samples may be more informative than others. We investigate this assumption in supervised image classification by biasing SGD (Stochastic Gradient Descent) to sample important samples more often during training of a classifier. In contrast to state-of-the-art, our approach does not require additional training iterations to estimate the sample importance, because it computes estimates once during training using the training prediction probabilities. In experiments, we see that our learning technique converges on par or faster in terms of training iterations and can achieve higher test accuracy compared to state-of-the-art, especially when datasets are not suitably balanced. Results suggest that sample importance has intrinsic balancing properties and that an importance weighted class distribution can converge faster than the usual balanced class distribution. Finally, in contrast to recent work, we find that sample importance is model dependent. Therefore, calculating sample importance during training, rather than in a pre-processing step, may be the only viable way to go.

## 1 INTRODUCTION

For many gradient-descent-based models increasing the model and training data sizes boosts the model performance and their ability to generalize. However, the increase in model and training data sizes comes with ever higher computational costs: longer training times and greater energy consumption are required to train a model on a given training data. One way to reduce these costs is to optimize the model training procedure. The most common training approach relies on random shuffling without replacement of the data samples during training for a given amount of epochs. As a consequence, all the samples are seen by the model the same number of times and are therefore implicitly treated as equally important.

Recent works on *hard example mining* (Felzenszwalb et al., 2009; Loshchilov & Hutter, 2015; Simpson, 2015; Alain et al., 2015; Shrivastava et al., 2016; Chang et al., 2017; Katharopoulos & Fleuret, 2018; Arriaga & Valdenegro-Toro, 2020; Pruthi et al., 2020; Lin et al., 2017) and *coresets selection* (Mirzasoleiman et al., 2020; Killamsetty et al., 2020; 2021; Yoon et al., 2021; Balles et al., 2022) have shown that the training samples are not equally important to learn a given task (Katharopoulos & Fleuret, 2018) and that it is possible to speed up training by respectively focusing on hard samples or subsets of the training dataset that best approximate the full gradient. These methods estimate sample importance during training in an online fashion and leverage this information to speedup the learning process. However, they often require an additional computational overhead to compute the sample importance, which makes them less effective in practice. This is particularly true in the case of *coresets selection* methods, which are based on conservative estimates (Paul et al., 2021).

Another research line has shown that the training samples in a dataset can be ranked according to different importance scores (Feldman & Zhang, 2020; Feldman, 2020; Jiang et al., 2020; Toneva et al., 2018; Paul et al., 2021) and that less important samples can be pruned prior training the model with little to no loss in test accuracy. Since computing sample scores exactly is computationally unfeasible, these methods rely on approximations that usually require to fully train at least one model, compute and rank the sample scores, and then select the smallest subset of samples best approximating the test accuracy achieved by a model trained on the full training dataset.

Our work is in between *hard example mining*, as we assume sample difficulty to be related to sample importance, and *data pruning*, as we compute sample importance only once after few training epochs, instead of adapting them during training as in *hard example mining* or *coresets selection*.

**Motivation** There is increasing interest in understanding the importance that training samples have on model generalization to unseen data, in order to boost the model performance by pruning, downweighting or downsampling less important samples. However, we find that existing methods generally face two main challenges. On one hand, they impose some computational overhead due to the computation of the samples' importance. On the other hand, they often rely on many parameters to select which samples to use during training to boost the model performance. We address these challenges by proposing a simple, yet effective learning method that allows a model to estimate sample importance early in training and to switch its focus on more important samples.

Our method is closely related to the data pruning method by Paul et al. (2021), which requires to train 10 model instances for 20 epochs to estimate the importance scores that are then used to train a new model instance from scratch. This method uses four parameters: two to compute the sample scores (the number of models and the training epochs) and two to slide a window over the scores to select the subset of samples that best generalizes on the test data (window size and position).

In contrast, our method does not require additional training iterations to compute sample importance, as it is computed only once using the predicted probabilities of a single model after a few training epochs. Then training continues by sampling important samples more often. Furthermore, our method has only two parameters: epoch $E$ at the end of which to compute the scores and a *focusing* parameter $\gamma$ (Lin et al., 2017), which we fixed as constant for all the experiments.

**Contribution** We propose an SGD-based method, which pretrains a model for few epochs under the assumption that all training samples are equally important for generalization, then it focuses on samples that are more important to learn the given task. In experiments we observe that

- our method can identify sample importance with reduced computational cost and converges faster to a comparable or even better solution compared to state-of-the-art approaches;
- sample importance evolves in a model-specific way during training;
- sample importance can automatically balance the class distributions, and sampling by class importance can be more beneficial than balancing the class distributions by number of samples;
- our method allows to use multiple augmentations more efficiently than the other baseline methods.

## 2  TRAINING OPTIMIZATION

Standard training of deep learning models makes the implicit assumption that each training sample is equally important in a uniform scan through all samples in every epoch. Each sample is used exactly the same number of times (i.e. the number of epochs). In contrast, we hypothesize that a sample can be more or less important for a given model to learn a given task, and that we can estimate sample importance during training to speed up learning by focusing on more important samples.

Our core idea is to capture sample importance as sample difficulty; the harder a sample the more a model can learn from it to improve generalization. On the other hand, a sample that can be easily classified can be considered already learned and, therefore, is less informative.

**Problem formulation** Let $S = (x_i, y_i)_{i=1}^M$ be a training dataset with $M$ samples, $V = (x_i, y_i)_{i=1}^L$ a test dataset with $L$ samples, where $x_i$ are images and $y_i$ one-hot encoded class labels. Let further $f_\theta(.)$ be a model (e.g. a neural network) that can be trained using a gradient descent optimizer (e.g. stochastic gradient descent, SGD), $\theta$ its parameters, $f_\theta(x_i)$ the output probabilities after *softmax* activation. Additionally, let $N$ be the number of training epochs, $B$ the training batch size, $T_e = \lfloor M/B \rfloor$ the number of batches (or iterations per epoch) and $T = T_e \cdot N$ the overall training iterations (i.e. mini-batch gradient updates). Moreover, let

$$P(i|S_e, S) = 1/(|S| - |S_e|)\mathbf{1}_{i \notin S_e} \tag{1}$$

be the standard training sampler that uniformly scans through all the samples at every epoch without replacement (Chang et al., 2017), where $\mathbf{1}$ is an indicator function and $S_e$ is the set of samples

already selected during the current epoch. Let $\mathcal{L}$ be a training loss function (here $\mathcal{L}(f_\theta(x_i), y_i) = -y_i \log(f_\theta(x_i)_{y_i})$, i.e. the Cross Entropy loss).

The gradient step at iteration $t$ can be computed as

$$\theta_t = \theta_{t-1} - \eta \frac{1}{B} \sum_{i=1}^{B} \nabla_{\theta_{t-1}} \mathcal{L}(f_{\theta_{t-1}}(x_i), y_i) \tag{2}$$

with learning rate $\eta$ and $(x_i, y_i)$ sampled from $S$ using the sampler $P(i|S_e, S)$. Furthermore, let $A(V, f_{\theta_t}(.))$ be the accuracy, i.e. the fraction of correctly predicted samples, of model $f_{\theta_t}$ after $t$ training iterations on all test samples $(x_i, y_i) \in V$. Then, let $A_b = \max_t(A(V, f_{\theta_t}(.)))$ be the highest test accuracy achieved by the model $f_\theta(.)$, and $t_b = \arg\max_t(A(V, f_{\theta_t}(.)))$ be the training iterations needed to reach $A_b$ using standard sampling (1). The 'speed-up factor' of a training method yielding model $f_{\theta'}(.)$ (note the $'$ indicating the other training method) is given as $t_b/t_n$, if it reaches baseline accuracy $A_b$ at $t_n$ iterations, given a tolerance $\epsilon$:

$$t_n = \min_{t \leq t_b}(t) \quad \text{s.t.} \quad A(V, f_{\theta'_t}(.)) \geq A_b - \epsilon \tag{3}$$

Informally, given a training and a test set, a model, an optimizer, and a baseline maximum accuracy, we search for the baseline maximum accuracy with the shortest amount of training iterations. Note that we choose accuracy as a generalization metric, but (3) can be extended to other metrics.

## 3   SGD WITH SAMPLE IMPORTANCE

We hypothesize that a model can learn a given task faster or more reliably by focusing on important samples in a given dataset. To find sample importance, we leverage the concept of focus during training used for Focal Loss (Lin et al., 2017), but from a different perspective: instead of regularizing the loss based on sample difficulty $(1 - f_\theta(x_i)_{y_i})$ (compare (5)), we define a sampler $P$ where we assign each training sample $(x_i, y_i)$ a weight $w_i$ of being selected according to its difficulty:

$$P(i|f_{\theta_E}, S) = \frac{w_i}{\sum_{j=1}^{M} w_j} \quad \text{with} \quad w_i = (1 - f_{\theta_E}(x_i)_{y_i})^\gamma, \quad \text{and} \quad (x_i, y_i) \in S \tag{4}$$

$P$ uses replacement and ensures that already learned samples (easy samples) are downsampled, whereas samples that are still to be learned (hard samples) are oversampled. $f_{\theta_E}(x_i)_{y_i}$ is the probability that model $f_{\theta_E}$ classifies the training sample $x_i$ in the correct class $y_i$, $f_{\theta_E}$ is the model pretrained using the uniform sampler without replacement (1) for $E$ epochs. Lastly, $\gamma$ is the *focusing* parameter (Lin et al., 2017), that regulates the focus to put on hard examples. Specifically, $\gamma > 0$ emphasizes hard samples, $\gamma < 0$ emphasizes easy samples and $\gamma = 0$ treats all training samples as equally important. Selecting the number of pretraining epochs $E$ appropriately is important, as too high $E$ reduces efficiency, and too small $E$ might lead to unreliable sample importance weights $w_i$. To select $E$ we take inspiration from the data pruning work Paul et al. (2021), where sample scores are computed after $E$ training epochs. However, differently from their work, we estimate sample importance using a single model instance (instead of 10) and do not train from scratch once the sample importance is obtained. By doing so, we ensure that no additional training iterations are required to estimate sample importance.

Furthermore, our method differs from other approaches that use samplers based on sample difficulty (Loshchilov & Hutter, 2015; Chang et al., 2017) since we use the sampler described in (4) and only compute sample importance once, after epoch $E$, then using them until the end of training. Pretraining techniques have already been used in other approaches to start updating the training sampler once the model predictions are stable and is also named *burn-in* (Chang et al., 2017), *warm-start* (Killamsetty et al., 2022; 2021) or *warm-up* (Katharopoulos & Fleuret, 2018).

We propose an SGD-based method biased towards important samples, which can be summarized as follows:

1. Pretrain a model $f$ on all training samples for $E \ll N$ epochs using the standard sampler (1).
2. At the end of epoch $E$, use the model predictions to update the sampler as described in (4).
3. Train $f$ for $N - E$ epochs using the new sampler, i.e. focusing on more important samples.

## 4 EXPERIMENTAL EVALUATION

We compare our sampling method against the following baselines and competing methods:

**Uniform sampling without replacement**   The standard deep learning pipeline trains a model by uniformly scanning through all the samples at every epoch. Formally, this translates into a training sampler according to (1). We denote this method as *SGD-Scan* (Chang et al., 2017).

**Focal Loss**   One approach to focus training to hard examples is Focal Loss (Lin et al., 2017), which adds a regularization term to the standard Cross Entropy loss

$$\mathcal{L}(f_\theta(x_i), y_i) = -\alpha_{y_i}(1 - f_\theta(x_i)_{y_i})^\gamma y_i \log(f_\theta(x_i)_{y_i}) \tag{5}$$

where $\alpha_{y_i}$ is the class weight for the class $y_i$, $(1 - f_\theta(x_i)_{y_i})^\gamma$ the regularization term used to emphasize hard examples. The same term is used as weights $w_i$ in (4). Note that here the loss of easy samples is downweighted, but all training samples are considered equally important from the point of view of the sampler. Our method does the opposite: it samples hard samples more often than easy samples, but uses the standard Cross Entropy as loss function. Additionally, our method computes sample importance once only (at the end of epoch E), whereas in Focal Loss it is computed in an online fashion (i.e. for every minibatch).

**Data pruning with EL2N scores**   Data pruning emphasizes training samples by ranking them according to an importance metric, pruning less important ones, and training a new model using a sampler (1) uniformly scanning through the remaining ones. We compare with the recent data pruning method by Paul et al. (2021), here denoted *EL2N*. This computes sample scores $s_i$ after $E = 20$ training epochs as average of $k = 10$ different model initializations, as $s_i = \mathbb{E}\,||f_\theta(x_i) - y_i||_2$, i.e. the expected $L_2$ norm of the error vector. Once the scores are ranked, $30\%$ of the worst scores are pruned for CIFAR10 and $10\%$ for CIFAR100, as these pruning percentages are reported as having best test performance (Paul et al., 2021).

**SGD sampled by prediction variance**   We also compare against a sampling method based on sample prediction variance (Chang et al., 2017), which emphasizes samples with high prediction variances with the following training sampler

$$P(i|H, S_e, S) \propto \left( \widehat{var}(f_{\theta,H_i^{t-1}}(x_i)) + \widehat{var}(f_{\theta,H_i^{t-1}}(x_i))^2/(|H_i^{t-1}| - 1) \right)^{1/2} + \epsilon \tag{6}$$

where prediction variance $\widehat{var}(f_{\theta,H_i^{t-1}}(x_i))$ is estimated on the history of stored prediction probabilities $H_i^{t-1}$ for sample $i$ until iteration $t - 1$, $\epsilon$ is a constant that prevents low variance instances from never being selected again. We denote this method *SGD-SPV*.

**SGD sampled by prioritization**   Lastly, we compare against the two sampling methods described in Schaul et al. (2015). The general sampling equation is $P(i|\delta, S) \propto p_i^\alpha / \sum_k p_k^\alpha$, where $\delta$ are the sample losses, and $p_i = |\delta_i| + \epsilon$ for *proportional prioritization* (here denoted *SGD-Prop*) and $p_i = 1/rank(i)$ for *rank-based prioritization* (here denoted *SGD-Rank*), where $rank(i)$ represents the rank of sample $i$ sorted according to $|\delta_i|$.

### 4.1 COMPARATIVE EVALUATION

We compare our method to the approaches above using CIFAR10, and CIFAR100, or variations by augmentations or omissions (see below and Appendix A for details). For our method, we set focus parameter $\gamma = 0.5$ for all experiments as preliminary experiments indicate it is a stable and performant parameter setting. For Focal Loss (5), we use $\gamma = 2$ as suggested in the original work (Lin et al., 2017) as best performing (further discussed in Appendix A.1.5).

Figure 1 shows test accuracy versus training iterations for ResNet50, trained on CIFAR10 and CIFAR100, respectively, for baseline and competitor approaches and our novel scheme. In both cases the spread between the maximum accuracies reached by the different training methods is $\approx 0.005$. Results are summarized in Table 1. For CIFAR10 (Figure 1), we observe that standard training SGD-Scan performs better than Focal Loss, on par with SGD-SPV and slightly worse than SGD-Prop, but

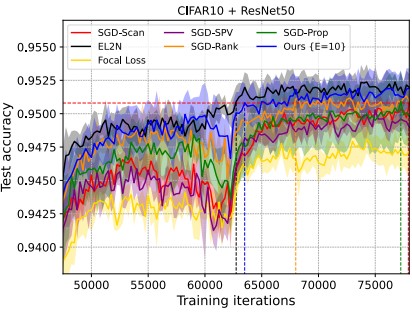 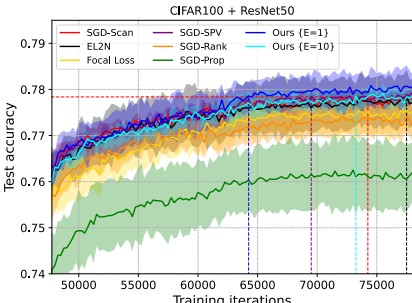

Figure 1: Test accuracy of ResNet50 on CIFAR10 (left) and CIFAR100 (right) of baselines, competitors and our approach with $\gamma = 0.5$ and variable $E$. Solid lines indicate the mean, shaded regions $16^{th}$ and $84^{th}$ percentiles of 4 independent runs (i.e. different random seeds). The horizontal red dashed line is average maximum accuracy by standard training (*SGD-Scan*), the vertical dashed lines the first iteration in which a method on average reaches that maximum accuracy.

Table 1: Maximum mean test accuracy achieved by all methods and respective standard deviation.

| Data | SGD-Scan | EL2N[1] | Focal Loss | SGD-SPV | SGD-Rank | SGD-Prop | Ours |
|---|---|---|---|---|---|---|---|
| CIFAR10 | $95.1 \pm 0.1$ | $\mathbf{95.2 \pm 0.1}$ | $94.8 \pm 0.1$ | $95.0 \pm 0.1$ | $95.1 \pm 0.1$ | $95.1 \pm 0.3$ | $\mathbf{95.2 \pm 0.2}$ |
| CIFAR100 | $77.8 \pm 0.2$ | $77.8 \pm 0.6$ | $77.5 \pm 0.6$ | $77.9 \pm 0.2$ | $77.4 \pm 0.4$ | $76.3 \pm 0.9$ | $\mathbf{78.1 \pm 0.4}$ [2] |

is clearly outperformed by SGD-Rank, EL2N and our approach. EL2N and our method achieve the same maximum performance, but since EL2N induces precomputation overhead, it is considerably slower than our method. For CIFAR100 the situation is different. On average, EL2N cannot improve against SGD-Scan, whereas our method can (in particular with E=1). Moreover, EL2N shows high variance over the different training runs (shaded regions in Figure 1). Consequently, while mean maximum performances of EL2N and Ours clearly differ, this is not the case for maximum performances over all runs. In conclusion our method reliably delivers on par or better performance than EL2N, and is faster when taking into account the precomputation required by EL2N.

In Figure 1, we can also see that the average peak accuracy $A_b$ of standard training SGD-Scan (red dashed horizontal line) is reached earlier by EL2N (black dashed vertical line) and Ours (blue dashed vertical line) for CIFAR10, and earlier (Ours) or later (EL2N) for CIFAR100. However, Figure 1 does not take into account the additional training iterations required to compute the *EL2N* scores, which are 7800 per model instance, for a total of 78000. We take this into account in Figure 2, where we investigate the speedup behaviour of the different methods and if it is possible to trade a slight loss of accuracy for additional speedup. Note that we only add 7800 training iterations to the overall iterations, instead of 78000, as the 10 model runs can be parallelized. However, in terms of energy consumption, those iterations are still additional overhead that our method does not require. In the figure, we show at which iteration each training method reached the maximum accuracy of standard training SGD-Scan minus a tolerance $\epsilon$; and derive the speedup factor as the quotient of the two iteration numbers $t_n/t_b$ (see (3)). In all cases our method yields top speedup for almost all $\epsilon$, only slightly surpassed by EL2N (dashed), but please recall that our figures ignore the overhead EL2N has for precomputing scores. Including the necessary pre-computations for EL2N in the analysis, our method is clearly faster.

## 4.2 SAMPLE IMPORTANCE IS MODEL DEPENDENT

Recent works on data pruning (Feldman & Zhang, 2020; Feldman, 2020; Jiang et al., 2020; Toneva et al., 2018; Paul et al., 2021) compute descriptive sample importance scores for a given dataset according to different metrics. Despite some suggestions on model and learning parameters dependency, no explicit empirical study is performed on model dependency of such scores. Paul et al.

---

[1]It requires 78000 additional iterations excluding hyperparameter tuning (the equivalent of an entire run).

[2]It refers to Ours E=1. We omitted Ours E=10 ($77.9 \pm 0.3$) for space limitations.

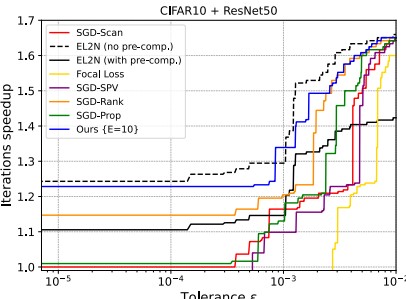 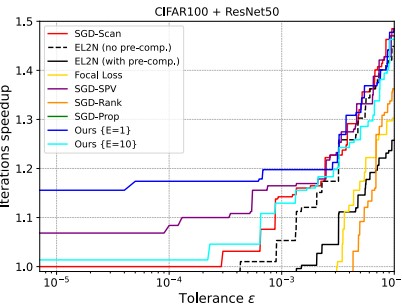

Figure 2: Speedup factor $t_b/t_n$ wrt. maximum accuracy achieved on average with *SGD-Scan*, relaxed with varying tolerance $\epsilon$ (from 0 to 0.01). Results of ResNet50 on CIFAR10 (left) and CIFAR100 (right) for baselines, competitors and our approach with $\gamma = 0.5$ and variable $E$.

(2021) even suggest that the scores are a property of the data, not of the model or task. By contrast, we provide empirical evidence that sample importance is not independent of model architecture (Toneva et al., 2018; Jiang et al., 2020). For assessment, we leverage the concept of *class importance* $\bar{w}^c$, aggregating sample importance by grouping sample weights $w_i$ (see (4)) by ground truth class $c = y_i$, i.e. $\bar{w}^c = \sum_i w_i|_{y_i=c} / \sum_{j=1}^M w_j$. We use model architectures with increasing number of parameters: LeNet (LeCun et al., 1989), PyramidNet110 (Han et al., 2017), GoogleNet (Szegedy et al., 2015), and ResNets 18, 34, 50, 101 (He et al., 2016) (more details in Appendix A.1.3). Figure 3 shows that class importance weights change depending on model architecture and size. However, importance weights derived from different models are correlated; the overall mean rank correlation is 0.77. E.g. we observe that all model architectures agree on the hardest (*cat*) and easiest (*automobile*) classes to learn, but GoogleNet for which *automobile* is second easiest after *ship*. Other classes are ranked differently depending on the model. For instance, the second hardest class is *dog* for PyramidNet110 and ResNet101 (and EL2N), *bird* for the other models. Class importance ranking induced by the scores agrees on the hardest classes, but not on the easiest one, which is *truck* before pruning and becomes *horse* after pruning.

Notice that here we report weights computed at the best tested epoch $E$ for each model. In Appendix B.1 we report weights computed at epoch $E = 10$ for every model, further showing that sample importance evolves in a model-specific way during training.

In other words, every model induces its own 'ideal' sample importance when learning a new task, but models agree to some extent. Therefore, it seem advisable that each model estimates its respective sample importance during learning and prioritize samples individually. As a consequence, precomputed sample importance scores may only be reliable when adopting the same (or at least a similar) model as the one used to compute them.

Note that by the same reasoning it is likely that sample importance also depends on the learning hyperparameters (optimizer, learning rate, schedules, etc.) as they shape model learning (Pruthi et al., 2020; Jiang et al., 2020), which we do not explore in this work. However we further discuss sample importance properties in Appendix B, among which task and initialization dependency.

### 4.3 SAMPLE IMPORTANCE HAS INTRINSIC BALANCING PROPERTIES

As mentioned at the end of Section 4.1, our method works best when there is a difficulty imbalance among the samples of a dataset. To further validate this hypothesis, we create such imbalance on CIFAR100 where 20% of the classes are randomly selected (once for all different runs) and augmented $A$ times (see Appendix A.1.1 for details).

In this setting, Figure 4 shows that our method converges faster and to a better solution with respect to the standard baseline. In particular, our method sacrifices a little accuracy on the augmented classes to focus more on the non-augmented ones, which contain less samples, therefore improving the accuracy on them (see also Appendix B.4). This suggests that our method automatically balances the class distributions by sampling more often samples that appear less, considering them more important for training. Note that we observe very similar results on ImageNet-1K (see Appendix C).

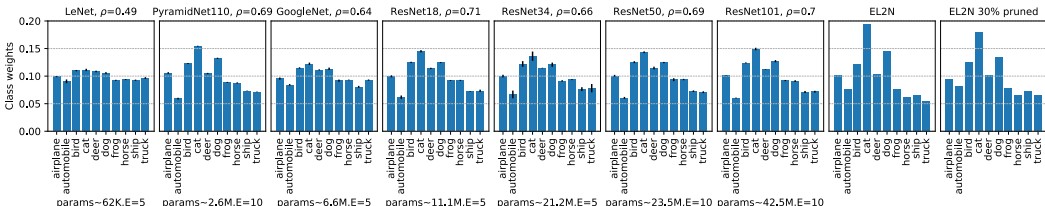

Figure 3: Class weights implicitly learned by LeNet, PyramidNet110, GoogleNet, ResNet18, ResNet34, ResNet50 and ResNet101 on CIFAR10 using our method for 4 different initializations (error bars) and class weights induced from EL2N scores before and after pruning 30% of the samples with lowest scores. Captions on top give average Spearman rank correlation $\rho$ of sample weights over the 4 initializations for the architectures, whereas those on bottom give the number of parameters and the epoch E.

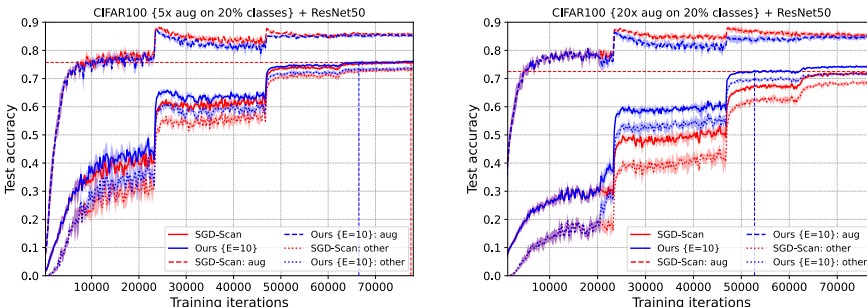

Figure 4: Test accuracy of all (solid lines), augmented (aug, dashed lines) and non-augmented classes (others, dotted lines) of ResNet50 on CIFAR100 using standard training (*SGD-Scan*) and our method, when the 20% of the classes are randomly selected and augmented 5 (left) and 20 (right) times.

**A balanced class distribution is not necessarily the best option** Methods like SMOTE (Chawla et al., 2002) try to balance the class distribution in tasks where class imbalance negatively affects model performance. We hypothesize that a balanced class distribution is not necessarily the best option and that classes may be weighted according to their importance for learning. In Figure 5 we compare the performance of (i) evenly balanced classes, i.e. a constant number of samples per class (SGD-Scan), (ii) our sampling method, and (iii) classes sampled according to the *class importance* weights (*SGD-SCI*, cmp. Figure 3) extracted from the sample weights computed by our method. For both datasets, SGD-SCI on average performs better than the baseline SGD-Scan. For CIFAR100, it performs slightly better than our per-sample importance sampling method in some iterations, but it converges to comparable results towards the end of training. Therefore, Figure 5 shows that it is possible to improve model performance on a given task by sampling based on *class importance*. However, we conclude that assigning weights or changing the number of samples in order to achieve an importance-balanced dataset should not be done on the basis of classes, but considering finer grained measures or statistics, here done per sample. Intuitively this makes sense, when considering that a class may consist of multiple complicated subclasses, that need some extra care in comparison to 'simple' classes.

### 4.4 SAMPLE IMPORTANCE ALLOWS TO EFFICIENTLY USE MULTIPLE DATA AUGMENTATIONS

Lastly, we hypothesize that our method can use multiple augmentations more efficiently than the other baselines, since it should be able to oversample the most important samples and downsample all the others. We therefore expect that our method can quickly identify which augmentations are most useful to learn the given task.

We test this by designing an experiment where the original dataset is augmented 5 times, resulting in a bigger training dataset (250000 training samples for both CIFAR10 and CIFAR100). Note that we

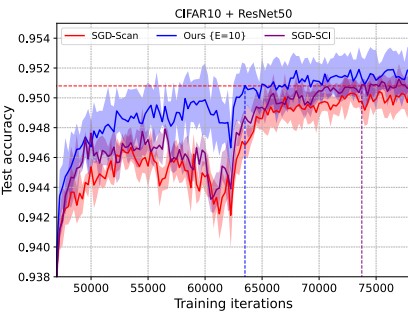 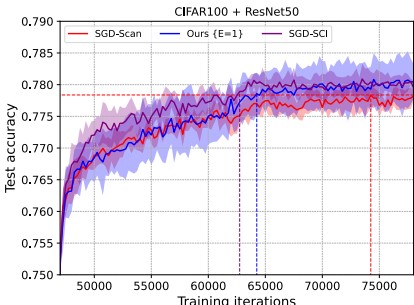

Figure 5: Test accuracy of ResNet50 on CIFAR10 (left) and CIFAR100 (right) using the baseline approach where classes are balanced (SGD-Scan), our sample importance approach with variable $E$ and $\gamma = 0.5$, and our class-importance sampled method (SGD-SCI).

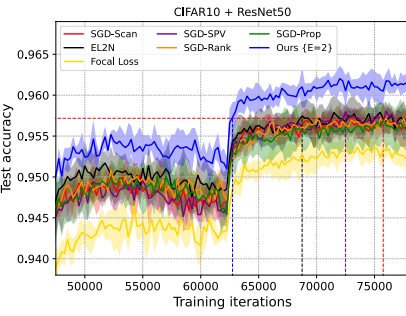 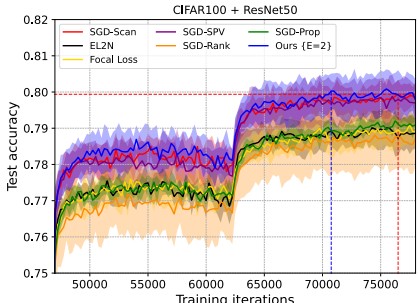

Figure 6: Test accuracy of ResNet50 on CIFAR10 (left) and CIFAR100 (right) using the baseline approaches and our method with variable $E$ and $\gamma = 0.5$ in presence of multiple augmentations.

still load the original images, but they are selected 5 times with 5 different augmentations (described in Appendix A) during training. We show the learning curves in Figure 6 and summarize the results in Table 2.

Comparing Figure 6 with Figure 1 shows that all methods strongly benefit from the augmentations. However, the spread between the maximum accuracies reached by the different methods approximately doubles. For CIFAR10 our method outperforms the other methods, and on CIFAR100 it is slightly better than SGD-Scan and SGD-SPV. Speedup is consistent with the results shown in Figure 1, suggesting that multiple data augmentation yield better results with approximately the same computational effort. Therefore using multiple augmentations can be beneficial as it comes at no additional cost in terms of iterations, but achieves a higher accuracy.

## 5 RELATED WORK

Our work relates to Focal Loss (Lin et al., 2017) in the use of focus and sample difficulty concepts, and to data pruning (Paul et al., 2021), which also aims to estimate sample importance. However, our method differs conceptually: we downsample less important samples instead of downweighting their loss, and we compute sample importance only once without training a new model from scratch.

Table 2: Maximum mean test accuracy achieved by all methods and respective standard deviation when using multiple augmentations. For the learning curves refer to Figure 6.

| Data | SGD-Scan | EL2N | Focal Loss | SGD-SPV | SGD-Rank | SGD-Prop | Ours |
|------|----------|------|------------|---------|----------|----------|------|
| CIFAR10 | $95.7 \pm 0.2$ | $95.8 \pm 0.2$ | $95.4 \pm 0.2$ | $95.8 \pm 0.1$ | $95.7 \pm 0.1$ | $95.7 \pm 0.3$ | $\mathbf{96.2 \pm 0.2}$ |
| CIFAR100 | $79.9 \pm 0.5$ | $79.0 \pm 0.5$ | $79.1 \pm 0.5$ | $79.9 \pm 0.8$ | $78.8 \pm 1.2$ | $79.2 \pm 0.2$ | $\mathbf{80.1 \pm 0.7}$ |

Other data pruning works suggest focusing on high influence and high memorization training samples (Feldman & Zhang, 2020; Feldman, 2020). Jiang et al. (2020) rank samples according to their structural regularities, while Toneva et al. (2018) suggest a forgetting score of the number of forgetting events a sample underwent during training. All of these methods suffer from the drawback that they have to compute sample importance first, prune accordingly and then train a new model from scratch. Moreover, this overhead increases considering that scores are likely model-specific (Toneva et al., 2018; Jiang et al., 2020), which implies that sample scoring needs to be repeated for every model and every dataset. Finally, an issue with these methods is that they require additional training iterations to compute sample importance scores, which our method avoids.

*Hard example mining* (Felzenszwalb et al., 2009; Loshchilov & Hutter, 2015; Simpson, 2015; Alain et al., 2015; Shrivastava et al., 2016; Chang et al., 2017; Katharopoulos & Fleuret, 2018; Arriaga & Valdenegro-Toro, 2020; Pruthi et al., 2020; Lin et al., 2017) tries to determine hard samples during training and use them to speedup convergence, under the assumption that hard samples are more informative for generalization. Exact methods to converge to the global optimum using hard (or important) samples are proposed for SVMs (Felzenszwalb et al., 2009) and MLPs (Alain et al., 2015). For deep neural networks, some works emphasize important samples after the backward pass, such as the Focal Loss (Lin et al., 2017), that downweights the loss of easy samples, or Online Hard Example Mining (OHEM) (Shrivastava et al., 2016), that only computes the backward pass for a subset of samples in a minibatch. Loshchilov & Hutter (2015); Simpson (2015) sample more often samples with the greatest loss in an online fashion. Other sampling strategies aim to reduce the variance of stochastic gradients during training (Katharopoulos & Fleuret, 2018) or to sample more often samples with higher prediction variance (Chang et al., 2017). Similarly, *coreset selection* methods (Mirzasoleiman et al., 2020; Killamsetty et al., 2020; 2021; Yoon et al., 2021; Balles et al., 2022; Killamsetty et al., 2022) try to find sample subsets during training approximating the gradient of the full training set. Coreset selection is also used to prevent catastrophic forgetting (Yoon et al., 2021; Balles et al., 2022) or to speedup hyperparameter tuning (Killamsetty et al., 2022). Coreset methods make conservative estimates and are found to be less effective in practice (Paul et al., 2021). Differently from these methods, we compute sample importance only once using (4), which emphasizes hard samples without adding the overhead required when estimating the sample importance in an online fashion.

## 6  DISCUSSION AND CONCLUSION

We investigated importance weighted sampling of training data in SGD-based learning of classifiers. Our method does not require any additional training iterations or overhead to estimate sample importance. Over all experiments it proved consistently at least on-par performance with competitors, but can achieve better results with less training effort, when burn-in epochs $E$ are well selected. Dependence on this additional parameter is the only limitation we observed.

Our results further show that sample importance weights are at least to some degree model specific. This suggests that alternative methods using pre-computed sample importance scores to prune the data might lead to non-optimal results, especially when the model used to train with the pruned data is different from the one used to compute sample importance.

We offer empirical studies on *class importance* showing class balancing properties of our method and suggesting that unlike commonly assumed, balancing classes by number of samples is not necessarily ideal. Our results instead show that a model trained with classes sampled according to *class importance* weights derived by our *sample importance* weights can outperform a model trained with balanced class weights. Still, we believe that fine-grained importance weights (i.e. per sample) are preferable as they allow the sampler to oversample the most important samples, independently from their class. Given these promising results, we hope to encourage further research on sample and class importance, to further understand the learning dynamics and optimize the training procedure.

Lastly, we show that our method makes a more efficient use of multiple augmentations, as it can select important samples from a larger pool of images. Thus, our method allows better results with approximately the same number of iterations as needed to fully train without augmentations. We see interesting opportunities to leverage our method also in applications where there is lack of data and where multiple augmentations could be key to achieve better performances. Here, our method allows the automatic focusing to those augmentations that actually benefit learning.

ETHICS STATEMENT

The main effect of our work is to speed up the learning process and automatically balancing classes. Speedup reduces energy consumption and potentially allows for using less powerful hardware for training, a tiny step towards democratizing AI. Balancing classes may help to emphasise underrepresented samples and thus may raise visibility of minorities in data. It very much depends on the target application if this results in highly desired fair treatment of minorities or an unfair deviation from underlying distributions.

REPRODUCIBILITY STATEMENT

All presented results are reproducible using our code, which will be published upon acceptance.

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

## A  IMPLEMENTATION DETAILS

### A.1  TRAINING DETAILS

#### A.1.1  DATA

We run our experiments on CIFAR10, CIFAR100 (Krizhevsky et al., 2009), all under the MIT licence. We use the datasets in the same way as they are used in the work by Paul et al. (2021): 50000 samples for training and 10000 for test when using CIFAR10 and CIFAR100. As data augmentation we pad all sides by 4 pixels, random crop to 32x32 pixels and horizontally flip the image with probability 0.5.

**Experiment on multiple data augmentation**  For the experiment where we adopt multiple data augmentations we use 5 data augmentations: twice the same data augmentations described above, AutoAugment (Cubuk et al., 2019) with CIFAR10 policy, RandAugment (Cubuk et al., 2020) and TrivialAugmentWide (Müller & Hutter, 2021) with default parameters.

**Experiment on balancing properties**  In the experiment showing the balancing properties, we randomly select 20% of CIFAR100 classes with a constant seed, then we use the data augmentations described above when augmenting 5 times, and we repeat those augmentation types 4 times when augmenting 20 times.

#### A.1.2  FRAMEWORKS

We implement our code using PyTorch (Paszke et al., 2017) and PyTorch Lightning (Falcon & The PyTorch Lightning team, 2019) as Deep Learning frameworks and Hydra (Yadan, 2019) to manage the configuration files. We take additional inspiration on how to combine these frameworks from existing templates[3][4].

---

[3]`https://github.com/ashleve/lightning-hydra-template`.
[4]`https://github.com/grok-ai/nn-template`.

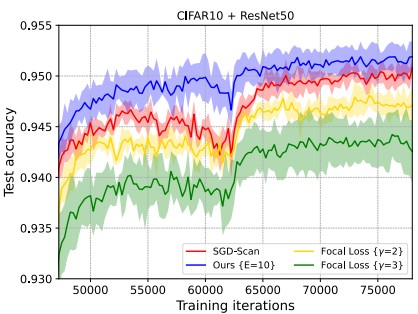 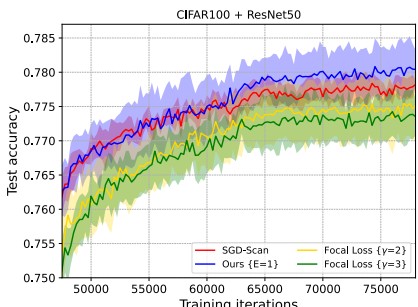

Figure 7: Different values of $\gamma$ in Focal Loss for CIFAR10 (left) and CIFAR100 (right). Note that $\gamma = 0.5$ leads to unstable results, therefore it is not reported.

### A.1.3  MODELS

We use ResNet50 (He et al., 2016) as main model. When working on CIFAR10 and CIFAR100, we apply the same changes applied by Paul et al. (2021): we replace the first two layers (convolution and max pooling) with a single convolution layer with 3x3 kernel and 1x1 stride. We also use ResNet18, ResNet34 and ResNet101 with these changes. Lastly, we use LeNet (LeCun et al., 1989), GoogleNet (Szegedy et al., 2015), PyramidNet110 and PyramidNet164 (Han et al., 2017).

Note that we use the models available in *torchvision* except for LeNet, which we adapt to our implementation from soapisnotfat (2017), and PyramidNet, which we take from dyhan0920 (2018).

### A.1.4  HYPERPARAMETERS

We use the same training hyperparameters used by Paul et al. (2021): Stochastic Gradient Descent (SGD) optimizer, initial learning rate = 0.1, Nesterov momentum = 0.9, weight decay = 0.00005. We use batch size = 128 for both CIFAR10 and CIFAR100. We train for 200 epochs, which translates into 78000 training iterations for both CIFAR10 and CIFAR100. We decay the learning rate after 60, 120 and 160 epochs, which for CIFAR10 and CIFAR100 translates to 23400, 46800, and 62400 iterations.

### A.1.5  BASELINES

We use the Focal Loss (Lin et al., 2017) implementation available at AdeelH (2020). In particular, we use $\gamma = 2$, as it yields the best results for both CIFAR10 and CIFAR100, as shown in Figure 7. Note that we additionally tested $\gamma = 3$ as suggested in Mukhoti et al. (2020), which leads to slightly worse results, and $\gamma = 0.5$, which leads to unstable convergence downgrading performance. For the EL2N method, we use the pre-computed scores provided to us by Paul et al. (2021) and pruned the data according to the percentages that yield the best performance in their paper. We use our own implementation for the other methods and adapt them to our pipeline. In particular, for SGD-SPV we use 10 *burn-in* epochs and $\epsilon = 0.2$.

### A.2  EXPERIMENTAL DETAILS

### A.2.1  REPORTING RESULTS

For every result, we report the mean, $16^{th}$ and $84^{th}$ percentiles of 4 independent runs (i.e. different random seeds). In the plots, this translates into a line (the mean) surrounded by a shading area (the percentiles). When showing the test accuracies over training, we zoom at the end of training to highlight the differences among the methods.

### A.2.2  TUNING THE METHOD'S PARAMETERS

Our method relies on 2 parameters: the amount of pre-training epochs $E$ and the *focusing* parameter $\gamma$.

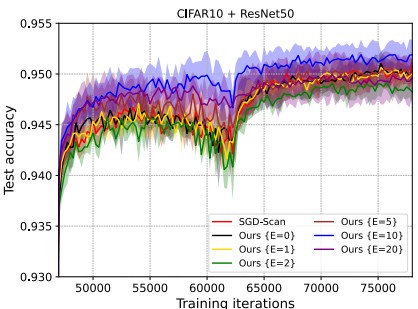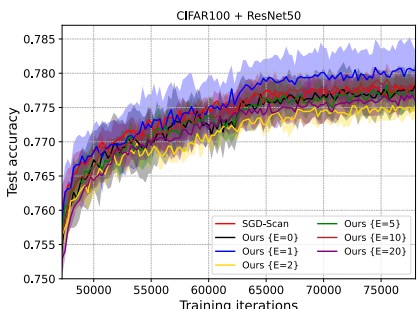

Figure 8: Sensitivity of $E$ when using our method with ResNet50 on CIFAR10 (left) and CIFAR100 (right) with $\gamma = 0.5$. The blue lines correspond to the results shown in the main paper.

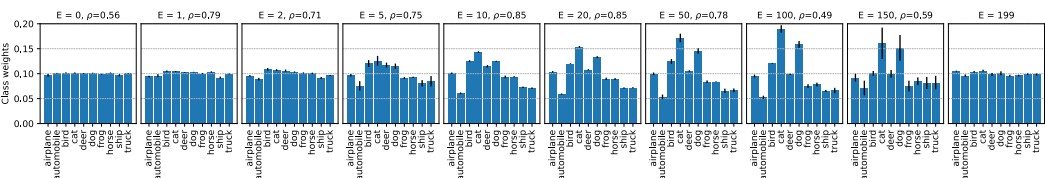

Figure 9: Class weights implicitly learned at different epochs by ResNet50 on CIFAR10 using our method for 4 different initializations (error bars). Captions on top give the epoch at which weights are computed and the Spearman rank correlation $\rho$ between the mean sample weights in a chart and the chart on its right (e.g. $\rho$ between mean E=0 and E=1 weights on the first chart, E=1 and E=2 on the second, and so on).

We start by finding a good value for the epoch $E$ at the end of which to compute the weights. We show that fewer than 20 pretraining epochs $E$ are enough to estimate sample importance and that going further decreases the generalization performance. We further show that $E$ depends on the dataset and on the model. In other words, a given (initialized) model seems to be able to quickly learn which samples are most useful to learn the given task, and by switching its focus on these samples it can converge faster.

Once a good epoch is found, we tune $\gamma$, which is used in (Equation 4) to convert sample difficulty to sample weights. We find that a good value for $\gamma$ is 0.5 and therefore we use this for all the experiments.

Unless differently specified, we use $E = 10$ for CIFAR10 and $E = 1$ for CIFAR100. Finally, for the experiments on CIFAR10 and CIFAR9 with LeNet, GoogleNet and ResNet18 we use $E = 5$.

**Sensitivity of $E$** Figure 8 shows that for $\gamma = 0.5$ our method is slighlty sensitive to the number of pretraining epochs $E$ with the uniform random sampler. For CIFAR10 we chose $E = 10$, but values close to 10 and between 5 and 20 can also be good candidates, whereas for CIFAR100 only $E = 1$ leads to better results than the baseline, and other values lead to comparable or slightly worse performance.

In Figure 9 we show how class importance weights (and therefore sample weights) change during training and how subsequent class weights are correlated. Sample weights computed after epoch 100 are less correlated than weights computed earlier in training.

**Sensitivity of $\gamma$** In Figure 10 we report the sensitivity analysis of $\gamma$ for the chosen values of $E$ for both CIFAR10 ($E = 10$) and CIFAR100 ($E = 1$). The figure shows that multiple values of $\gamma$ can be used for the same value of $E$ and achieve comparable results. For instance, in CIFAR10 both $\gamma = 0.5$ and $\gamma = 1$ achieve better performance than the baseline, whereas only $\gamma >= 2$ lead to drop in performance. For CIFAR100, on the other hand, multiple values of $\gamma$ lead to better performance than the baseline (0.1, 0.5, 1.5, 3, whereas the other tested values lead to comparable or slightly worse results.

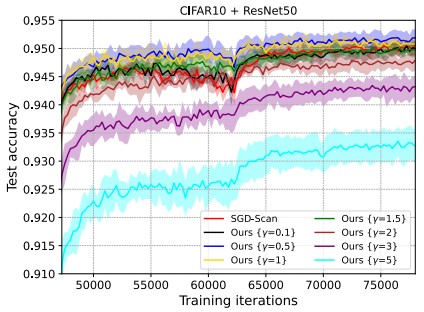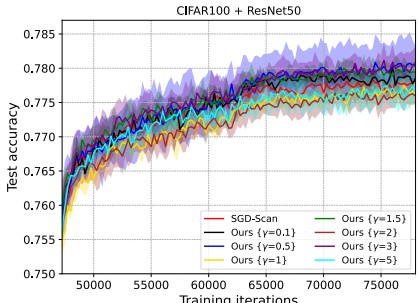

Figure 10: Sensitivity of $\gamma$ when using our method with ResNet50 on CIFAR10 with $E = 10$ (left) and CIFAR100 with $E = 1$ (right). The blue lines correspond to the results shown in the main paper.

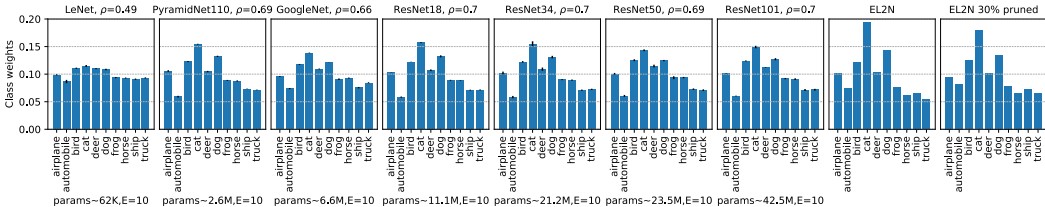

Figure 11: Class weights implicitly learned by LeNet, PyramidNet110, GoogleNet, ResNet18, ResNet34, ResNet50 and ResNet101 on CIFAR10 using our method for 4 different initializations (error bars) and class weights induced from EL2N scores before and after pruning $30\%$ of the samples with lowest scores. Captions on top give average Spearman rank correlation $\rho$ of sample weights over the 4 initializations for the architectures, whereas those on bottom give the amount of parameters and the epoch E.

### A.2.3 COMPUTING RESOURCES

We run all experiments on a single NVIDIA A100-SXM4-40GB, which is provided by an internal cluster. We use approximately 5000 GPU hours for the whole project. The experiments we show in the main paper and Appendix took approximately 700 GPU hours to compute.

## B FURTHER ANALYSIS ON SAMPLE IMPORTANCE

### B.1 MODEL DEPENDENCY

Figure 11 reports class importance weights aggregated from sample importance weights computed by our method with different models and parameters $E = 10$, $\gamma = 0.5$. Comparing this figure to Figure 3 shows that class weights from different models get closer when computed at the same epoch (the mean rank correlation of sample weights from all models is 0.81), but there are still differences between class weights, implying that sample importance evolves in a model-specific way during training.

Figure 12 (left) reports several ablation studies to better show the model dependency of sample importance. In particular, it shows that using weights computed by LeNet instead of those computed by our method at epoch $E = 10$ when training a ResNet50 on CIFAR10 leads to worse results (comparable to the main baseline, SGD-Scan) than using the weights computed by ResNet50. This further shows that sample weights are model-dependent.

Note that using weights computed by ResNet18 at epoch $E = 5$ instead of those computed by our method at epoch $E = 10$ with ResNet50 leads to slightly better results, whereas using weights computed by ResNet18 at epoch $E = 10$ leads to slightly worse results. We hypothesize that ResNet18 and ResNet50 share some network similarities and therefore cannot be considered as completely different models, however additional experiments are required to verify this hypothesis as well as the possibility to use weights computed by a smaller ResNet into a bigger one.

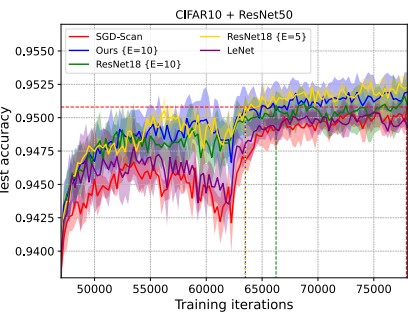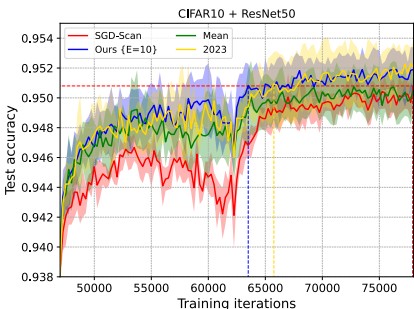

Figure 12: Test accuracy of ResNet50 on CIFAR10 using our method with $E = 10$, but using other weights instead of computing them, as explained in the following. In both Figures, SGD-Scan and Ours E=10 are respectively the uniform sampling and our method with $E = 10$. (Left) *ResNet18 $E = 5$* and *ResNet18 $E = 10$* respectively use the weights computed by ResNet18 with the same initialization at the specified epochs, *LeNet* uses weights computed by LeNet at epoch $E = 5$. (Right) *Mean* uses the average sample weights over the 4 runs computed with ResNet50, *2023* uses sample weights computed with ResNet50 when using seed=2023 (seed used for the last run).

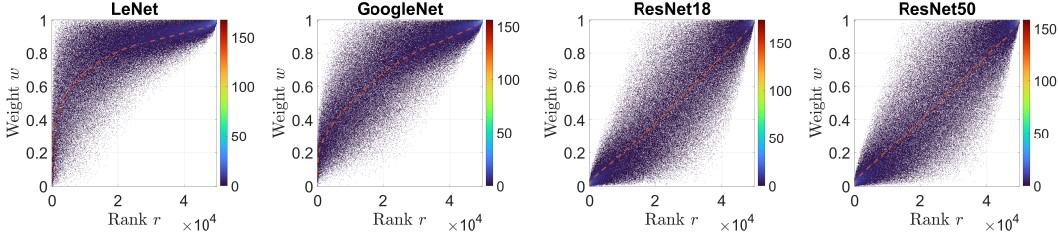

Figure 13: Sample weights $w_i$ over four different initializations for four different models trained on CIFAR10. Samples are sorted by their mean assigned weight $\bar{w}_i$. The x-axis is the resulting rank $r \in \{1, ..., M\}$ ($M = 50k$ for CIFAR10) from lowest $\bar{w}$ to highest $\bar{w}$. The dashed red line indicates $\bar{w}$. In case plotted points overlap, color changes according to the number of overlapping points.

## B.2 INITIALIZATION DEPENDENCY

Figure 13 shows initialization dependency with ResNet50 on CIFAR10 for 4 runs of our model with different initializations each. For each run, the resulting weights $w_{i,j}$ are averaged $\bar{w}_i = \sum_{j=1}^{4} w_{i,j}$, where $j \in \{1, ..., 4\}$ indicates the runs. Samples are sorted in ascending order of $\bar{w}_i$ with $r(i)$ their resulting rank, i.e. the resulting new index $r(i) \in \{1, ..., M\}$. The plots show $(r(i), \bar{w}_{r(i)})$ as dashed red line and the dark blue dots are $(r(i), w_{r(i),j})$ with $j \in \{1, ..., 4\}$. So for each sample with index $r(i)$ we see the initialization dependence of weights $w_{r(i),\cdot}$ as vertical spread of the blue points. We observe that for all models the variation in $w_{r(i),\cdot}$ is small for the hardest samples with $r(i) \lesssim M$ and all assigned weights $w_{r(i),\cdot} \gtrsim 1 - \epsilon$, with some acceptable threshold $\epsilon$, say $\epsilon = 0.1$. These samples are consistently detected as hard and could be identified reliably in a pre-processing step. The same is true for the easiest samples $r(i) \gtrsim 1$ and $w_{r(i),\cdot} \lesssim \epsilon$, but less pronounced for the small networks LeNet and GoogleNet. For medium hard samples, i.e. samples with $\bar{w}_{r(i)} \approx 0.5$ the variation of $w_{r(i),\cdot}$ is much higher. In many cases the same sample may be assigned $w_{r(i)} \ll 0.2$ in one initialization and $w_{r(i)} \gg 0.8$ in another initialization. Thus, depending on the initialization, a sample may be easy or hard for the same model.

Figure 12 (right) shows the test accuracies of ResNet50 on CIFAR10 with two additional variants of our method, compared to uniform sampling and our approach with $E = 10$. Both variants start training a ResNet50 as our method until epoch $E = 10$, then update the sample weights as: the average sample weights (*Mean*) computed when using our approach with ResNet50 on 4 different runs, and sample weights computed with seed 2023 (fourth run) for all the runs. The figure shows that using the average sample weights leads to slightly worse results, whereas using sample weights

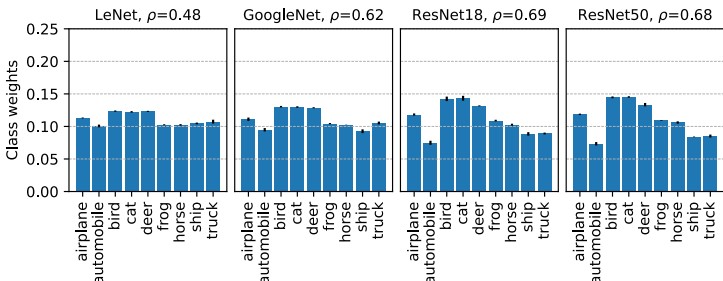

Figure 14: Class weights implicitly learned by ResNet18 and ResNet50 using our method for 4 different initializations (error bars) on *CIFAR9*, a subset of CIFAR10 without class *dog*. As for Figure 3, captions report average Spearman rank correlation $\rho$ of sample weights over the initializations.

computed by a different seed (except for the last run with seed 2023, which uses its original sample weights) can lead to comparable results. This suggests that there is an initialization dependency, but it is not strong.

We conclude, that sample difficulty has a minor dependence on initialization, when computed from model output $f_\theta(x_i)_{y_i}$. Consequently, it may not necessarily be possible to pre-compute (i) sample difficulty, (ii) an optimal selection of samples, or (iii) an optimal choice of per sample weights. It may be necessary that weight assignment is done during training, as proposed here.

### B.3 TASK DEPENDECNY

We show the task dependency by running our method on a subset of CIFAR10, where the third most confused class (*dog*) is removed. We denote this subset *CIFAR9*. Not surprisingly, the figure shows that the class weights slightly change together with the task, although some general agreement on the class importance is retained. Of particular interest are the classes *bird* and *cat* in ResNet18, which in *CIFAR9* appear to be very close in difficulty, in contrast to the class weights in CIFAR10 (Figure 3), where it is evident that class *cat* is harder than *bird*.

Note that for this experiment we adjust the training iterations according to the data size (45000) and therefore we run 70200 iterations. Here the learning rate is decayed after 21060, 42120, 56160 iterations, that is (60, 120, 160 epochs) × 351 iterations per epoch.

### B.4 CLASS WEIGHTS IN PRESENCE OF CLASS IMBALANCE

Figure 15 shows the class importance weights computed by our method with $E = 10$ and $\gamma = 0.5$ using ResNet50 on CIFAR100 with 20% of the classes augmented 5 times (top) and 20 times (bottom) (corresponding to Figure 4 left and right). Consistently with our assumption we see that class weights are reduced for augmented classes, where weights get lower when using more augmentations.

## C EVALUATION ON IMAGENET-1K

We additionally evaluate our method on ImageNet-1K (Russakovsky et al., 2015) using a vanilla ResNet50 with a slight modification of the PyTorch procedure described in Wightman et al. (2021), where the learning rate scheduler is decayed after 30, 60 and 75 epochs, because we noticed that decaying the learning rate once more towards the end slightly improves the final accuracy (i.e. 76.4% instead of 76.1%). Even though we did not run an extensive search to tune our hyperparameters, Figure 16 shows that using our method with E=10 and $\gamma = 0.5$ achieves slightly worse results towards the end of training, whereas with $\gamma = 0.1$ achieves comparable performance to standard training. However, from Figure 17, where we augment 20% of the classes (randomly selected once) 20 times, we conclude that our method works best when there is a difficulty imbalance in the given dataset and that, when samples are (almost) equally difficult, it performs on par with vanilla SGD.

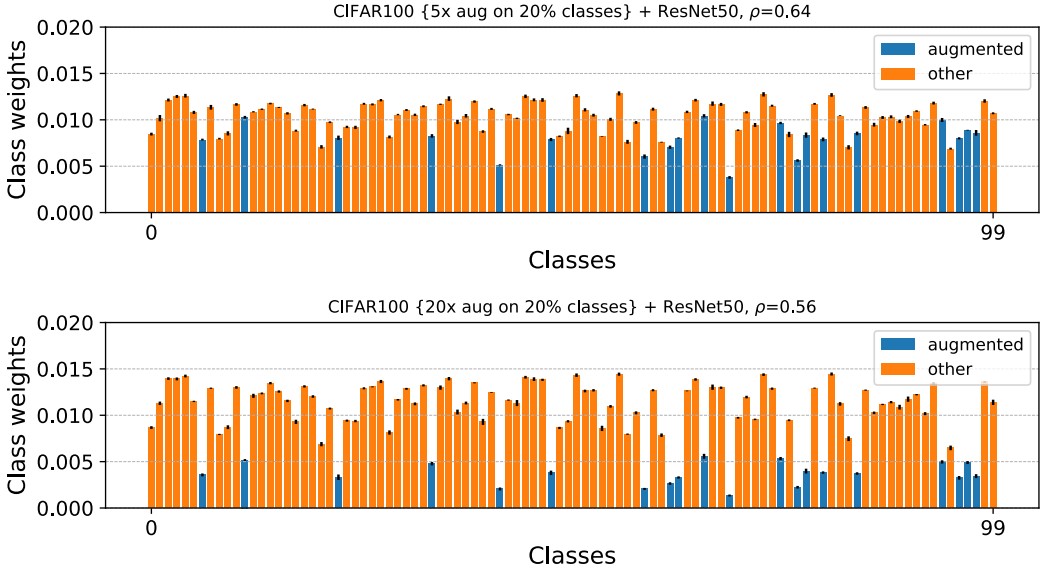

Figure 15: Class weights implicitly learned by ResNet50 on CIFAR100 where 20% of the classes are randomly selected once and augmented 5 times (top) or 20 times (bottom). Error bars show the standard error over 4 different initializations. Caption on top give average Spearman rank correlation $\rho$ of per sample weights over the 4 initializations. Non-augmented classes are shown as *other*.

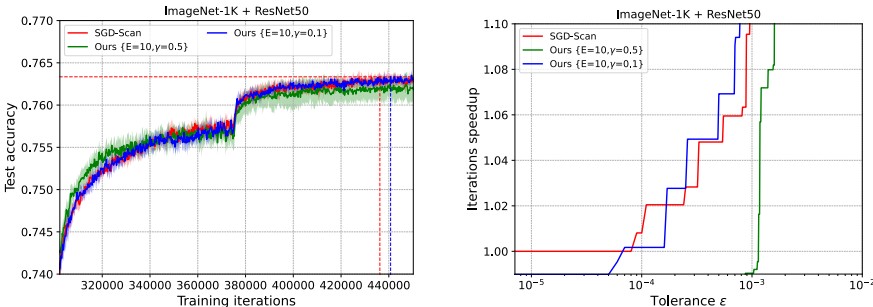

Figure 16: Test accuracy (left) and speedup in terms of iterations (right) of ResNet50 on ImageNet-1K using standard training (*SGD-Scan*) and our method.

The figure shows that using our sample weights leads to better results than using random weights and than focusing on easy samples, computed at the same epoch as described above. Moreover, the disentangled variant achieves slightly worse results than uniform sampling, further suggesting that showing more often samples that are more important using their full gradient leads to an better performance.

**Augmentation details**   For the experiment in Figure 17 we use the following set of 5 different augmentations repeated 4 times, where all sets are preceded by RandomResizeCrop with size 224: RandomHorizontalFlip, RandomHorizontalFlip and ColorJitter, AutoAugment (Cubuk et al., 2019), RandAugment[5] with parameters suggested in Wightman et al. (2021) for procedure A3, and Trivial-AugmentWide (Müller & Hutter, 2021).

---

[5]from `https://github.com/rwightman/pytorch-image-models`

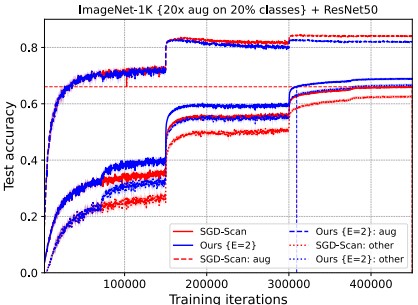 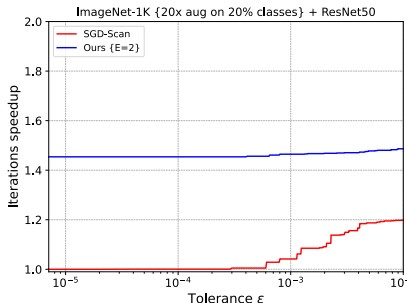

Figure 17: (Left) Test accuracy of all (solid lines), augmented (aug, dashed lines) and non-augmented classes (others, dotted lines) of ResNet50 on ImageNet-1K using standard training (*SGD-Scan*) and our method, when 20% of the classes are randomly selected and augmented 20 times. (Right) Speedup in terms of iterations of SGD-Scan and our method with respect to the maximum accuracy achieved by SGD-Scan.

## D    VISUAL INSPECTION

In Figure 18 and Figure 19 we visualize respectively the easiest and hardest sample per class (columns) and per seed (rows). These figures reveal different interesting observations:

- often the difference between easier and harder is the different orientation of the object inside the image (e.g. the frontal cat is considered easy, whereas cat seen from other angles is harder, perhaps because there are less images like so)

- it is clear that our method succeeds in identifying easy and hard samples

- some of the hardest samples are outliers (e.g. the first 2 images from top of the truck column (right-most).

- there is some disagreement in sample importance between the different initializations. This further confirms our initialization dependency claim.

## E    ADDITIONAL EMPIRICAL EVALUATIONS

Lastly, in the following subsections we show additional ablations in order to give further motivation for our method.

### E.1    COMPUTING WEIGHTS ONLINE AND SGD WITH REPLACEMENT

To the best of our knowledge, we are the first to empirically show that it is possible to estimate sample importance once early in training and to achieve better results by continuing training using this estimate as sampling distribution.

In Figure 20, on the other hand, we show that using our method in an online fashion (i.e. updating the sampling distribution after every iteration) leads to worse results. The figure also shows the results for SGD with replacement.

### E.2    IMPORTANCE METRIC CHOICE

We selected (4) as importance metric as in experiments it proved to be better than other importance metrics (loss, EL2N Paul et al. (2021), gradient norms), as shown in Figure 21.

Note that to compute the gradient norms with our current framework we had to loop over each minibatch after computing the per-sample losses to compute the per-sample gradient. This procedure took approximately 35 minutes for the single epoch E. Doing this online would be prohibitive.

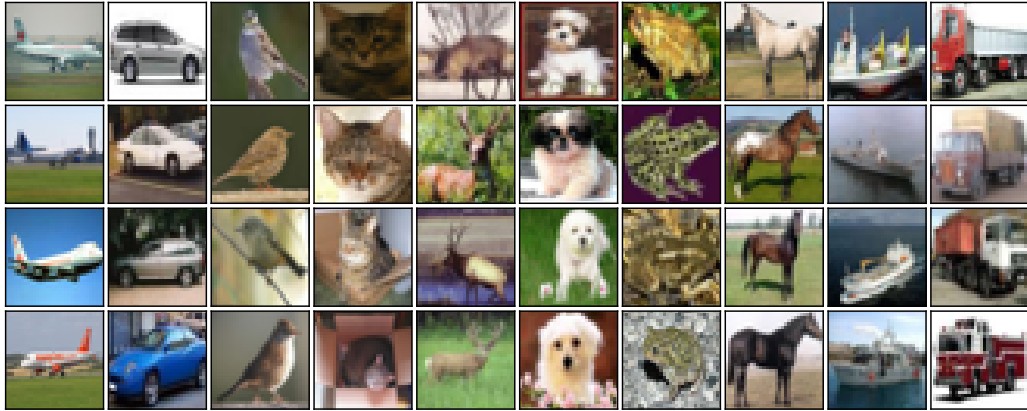

Figure 18: Sample with the lowest weight per class (columns) for 4 different initializations (rows) of ResNet50 on CIFAR10 with $E = 10$.

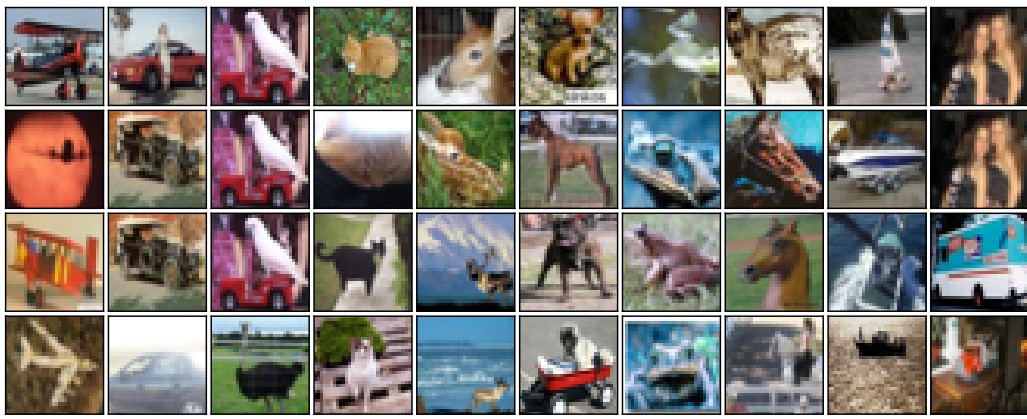

Figure 19: Sample with the highest weight per class (columns) for 4 different initializations (rows) of ResNet50 on CIFAR10 with $E = 10$.

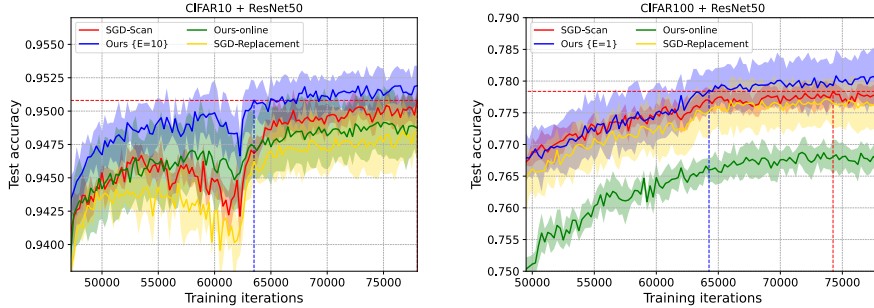

Figure 20: Test accuracy of ResNet50 on CIFAR10 (left) and CIFAR100 (right) of SGD-Scan, our method, our method with sample importance distribution updated online, and SGD with replacement.

### E.3 BIASED VS UNBIASED SAMPLING

We propose an SGD-based method biased towards important samples, as we assume that the given dataset is biased (i.e. it has an imbalance in sample importance). For comparison, in Figure 22 we report results for an unbiased version of our method, where we weight the gradient steps by the

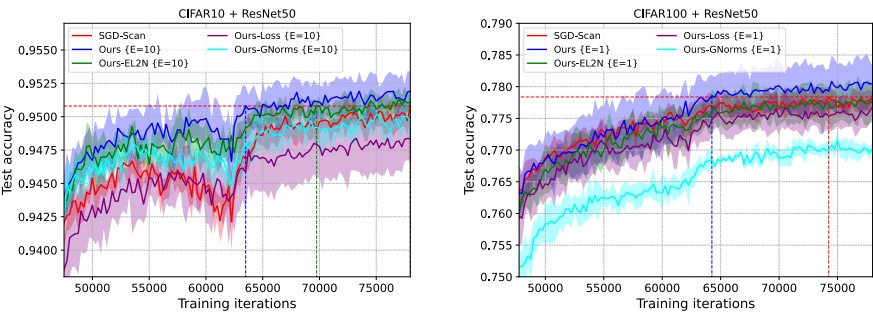

Figure 21: Test accuracy of ResNet50 on CIFAR10 (left) and CIFAR100 (right) of SGD-Scan, our method, our method with differnt importance metrics: EL2N (without ensembling), loss, and gradient norms (GNorms).

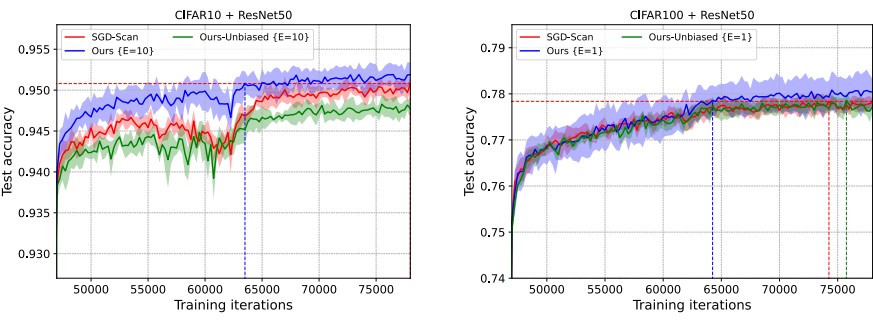

Figure 22

inverse of our sample weights (4). The figure shows that using a biased version of our method leads to better results.

### E.4  SAMPLING PROPORTIONAL TO EL2N SCORES

Lastly, Figure 23 shows the results for a method that samples proportional to the provided EL2N scores (instead of pruning and using SGD-Scan on the pruned data). The experiment shows that EL2N scores used as sample weights don't perform as good as the same scores used for data pruning and then training using SGD-Scan on the resulting pruned dataset.

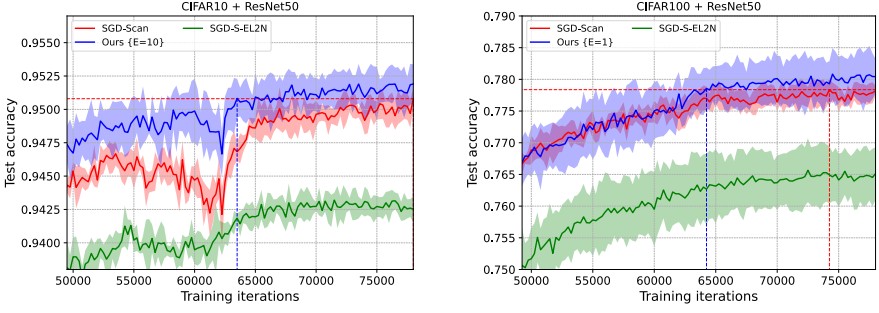

Figure 23: Test accuracy of ResNet50 on CIFAR10 (left) and CIFAR100 (right) of SGD-Scan, our method, and SGD sampled proportional to the provided EL2N scores (without pruning).

