# OpenReview forum: "Sample Importance in SGD Training"
_ICLR.cc/2023/Conference — Submitted to ICLR 2023_

### Official Review · Reviewer_VNC6 · 2022-10-23

**Confidence:** 4
**Correctness:** 2
**Technical Novelty And Significance:** 2
**Empirical Novelty And Significance:** 2
**Recommendation:** 3

**Clarity, Quality, Novelty And Reproducibility:**

Please see the Weaknesses section above for comments on clarity, quality, and novelty.

**Strength And Weaknesses:**

## Strengths
* The problem of speeding up training time of models is highly relevant and significant in the ML community
* Some components of the method are contextualized with respect to related work (e.g, Focal Loss, Lin et al., 2017)

## Weaknesses
* There is a significant amount of relevant prior work that is not appropriately discussed or cited. For instance, prior work is rich with theoretically-grounded methods for SGD Importance Sampling [1-3]. In fact, a closed-form solution for the importance sampling distribution that minimizes the sampling variance is known (sample proportional to gradient norms) [1].
* Unlike the prior works mentioned above, the proposed approach seems to arbitrarily pick the Focal Loss’ sample difficult criterion with $\gamma = 0.5$ with no compelling justification. In this regard, the work also overlooks deep connections with prior work. For instance, the EL2N score is an approximation of the (expected) gradient norms (see Paul et al., 2021, Definition 2.3 [4]), so sampling proportionally to EL2N scores (with respect to the most-up-to-date model) can be viewed as approximating the closed-form optimal importance sampling distribution [1]. What’s more is that the Focal Loss with $\gamma = 0.5$ is closely related to EL2N score evaluated for the same model (square root of EL2N score upper bounds Focal Loss with $\gamma = 0.5$).
* The authors claim that a main benefit of this work is that the hard points are identified using a model snapshot after training with standard SGD for E epochs. But it is not clear why the updated values for the model’s predictions (even if they are from previous iterations) are not used to adaptively update the importance sampling distribution every once in a while. Additionally, can't the same procedure be applied but with EL2N scores instead?
* How does the method compare to sampling proportional to EL2N scores instead? It’s not clear whether the choice of the Focal Loss metric is essential for the proposed method. It is also not clear how the reweighting of the selected examples are done. Relatedly, since the samples are no longer picked uniformly at random, each sample has to be reweighted by the sample probability so that the entire data set’s loss is captured in expectation (see [1-3]), but my understanding is that this is not implemented as part of the method.
* The method uses a previously studied importance metric (Focal Loss) to sample points, without adequate justification or substantial novelty relative to prior work.
* The empirical evaluations are not statistically significant or compelling. For instance, in Fig. 1 and Table 1 the performance of the proposed approach is well within one standard deviation of the approach based on EL2N scores.


[1] https://arxiv.org/pdf/1803.00942.pdf

[2] https://proceedings.neurips.cc/paper/2018/file/967990de5b3eac7b87d49a13c6834978-Paper.pdf

[3] https://arxiv.org/pdf/1401.2753.pdf

[4] https://openreview.net/pdf?id=Uj7pF-D-YvT


**Summary Of The Paper:**

This paper presents a data sampling approach to reduce the training time by focusing on difficult points. The main idea is to modify SGD so that it samples points with probability proportional to their sample difficulty, which the authors define using a score that is similar to the EL2N score of Paul et al. (2021), rather than uniformly at random as in standard SGD. The probability of sampling each point is computed once after training the model for E epochs using standard SGD, where E is a small number. Empirical results of the model’s test accuracy as a function of the training iteration, among other metrics, are compared to other baseline methods.

**Summary Of The Review:**

The authors tackle a problem that is very relevant and of high significance to the ML community. However, the work is not well placed in the context of prior works that have addressed importance sampling for SGD and fundamental connections are not discussed. The method is not clearly described, seems ad-hoc, and leaves the reader with a lot of questions about its connections to related work (why not sample according to EL2N scores computed over 1 model, which has some theoretical justification, for example?). The empirical evaluations are also not convincing and the purported gains are not statistically significant. In light of these considerations, I recommend rejection.

---

### Official Review · Reviewer_LLpx · 2022-10-25

**Confidence:** 4
**Correctness:** 3
**Technical Novelty And Significance:** 3
**Empirical Novelty And Significance:** 2
**Recommendation:** 5

**Clarity, Quality, Novelty And Reproducibility:**

- Quality: The paper overall is in reasonable quality.

- Clarity: The paper is sufficiently well-written. Definition and notation is clearly presented and sufficiently to be understood.

- Originality: The difference of the work compared to prior ones is well presented in the writing sections. However, the margin on empirical performance is limited.

- Reproducibility: reasonable.

**Strength And Weaknesses:**

[Strength]
1. The proposed sampler idea is simple and clear to be conducted. It uses the early trained model to produce sample importance and thus no additional training iteration is needed.
2. The empirical analysis on model dependence and intrinsic balancing properties is interesting.

[Weaknesses]
1. My  main concern lies in experimental results.
- The performance on CIFAR-10 by different methods make very little difference.  Numbers in bold should be those are significantly improved without overlap, which is not the case of CIFAR-10. This makes the comparison on CIFAR-10 less interesting.
- The performance gap on CIFAR-100, besides SGD-Prop, is still very marginal. The "Ours" in Table 1 misses the hyperparameter of E=1 or E=10. The Ours with E=10 achieves inferior performance from Figure 2. The option on E seems to make significant difference on the overall performance.
- Training on CIFAR-like-size dataset would be less insterested regarding saving the training cost. Experiments on larger dataset would be more persuasive.

2. The maximum mean test accuray seem to make a good saling point for the proposed method that it achieves the *best* test performance faster. Whereas this *best* performance could not be defined as the best until all the pre-defined iteration is finished. Thus, this faster speed-up seems to be less attractive in practice.

3. I wonder whether there could be some discussion on extending such sampler with other optimizer (besides the dependence as in sec 4.2), e.g. Adam. This coud make the proposed method more useful to a wider community.


**Summary Of The Paper:**

The authors propose a new sampling method for training deep neural network. The sampler is based on the sample importance, which is defined as the sample difficulty using the propability. This probability is produced by the early trained model using standard uniform sampling. The following training epoch will use the new sampler thus focus on more important samples. The proposed method is claimed to converge faster and achieve better test accuracy.

**Summary Of The Review:**

Overall, I think this paper proposes an interesting sampler idea that requires no additional training iteration, which saves computational cost compare to one exisiting data pruning method. However, the empircal results show marginal benefit compared to other methods, which would limit the

---

### Official Review · Reviewer_cDss · 2022-10-26

**Confidence:** 4
**Clarity, Quality, Novelty And Reproducibility:** Look good to me
**Correctness:** 3
**Technical Novelty And Significance:** 2
**Empirical Novelty And Significance:** 2
**Recommendation:** 3

**Strength And Weaknesses:**

Strength
1. The proposed method is explained clearly and with reasonable motivation.
2. Extensive experiments are conducted.


Weaknesses
1. The key idea of the proposed method is to use the first E epochs as the warmup stage to obtain importance scores. This idea is quite straightforward and its variants have been considered in several existing works (e.g., Chang et al., 2017; Killamsetty et al., 2022; 2021; Katharopoulos & Fleuret, 2018). It’s not clear why the proposed method is better than these existing methods. Actually, the authors mentioned these methods in the paper but didn’t compare against them in the experiments.
2. In addition, there are many recent works which take into account example priority for SGD-like methods (for example, see reference [1-4] below). The authors have considered Schaul et al. (2015) in the experiments. But to make the experiments more convincing, more recent priority-based methods should be considered and compared with. Otherwise, readers may feel the baseline is not state-of-the-art.
4. The paper may benefit from some in-depth analysis about the proposed method. For example, is there any reason why the proposed method should be better than baselines? Do we expect better performance under all scenarios? Other similar works usually contain this kind of analysis (see reference [1-4] below).

Reference:
[1] Katharopoulos, Angelos, and François Fleuret. "Not all samples are created equal: Deep learning with importance sampling." International conference on machine learning. PMLR, 2018.
[2] Hacohen, Guy, and Daphna Weinshall. "On the power of curriculum learning in training deep networks." International Conference on Machine Learning. PMLR, 2019.
[3] Liu, Rui, Tianyi Wu, and Barzan Mozafari. "Adam with bandit sampling for deep learning." Advances in Neural Information Processing Systems 33 (2020): 5393-5404.
[4] El Hanchi, Ayoub, David Stephens, and Chris Maddison. "Stochastic Reweighted Gradient Descent." International Conference on Machine Learning. PMLR, 2022.


**Summary Of The Paper:**

This paper proposed a sampling method for SGD that takes into account the importance of training examples. Specifically, the proposed method trains using uniform sampling for the first E epochs, and then compute importance scores for all training examples based on how accurate the current model’s prediction is. The proposed method trains using weighted sampling based on the importance scores for the rest epochs.


**Summary Of The Review:**

This proposed method looks reasonable, but its idea is not new. In addition, this paper lacks enough analysis and the experiments can be improved with more recent and relevant baselines.

---

### Official Review · Reviewer_auZ4 · 2022-10-26

**Confidence:** 4
**Correctness:** 3
**Technical Novelty And Significance:** 3
**Empirical Novelty And Significance:** 1
**Recommendation:** 5

**Clarity, Quality, Novelty And Reproducibility:**

The writing is clear and the experiments shown are simple, so potentially easily reproducible.

**Strength And Weaknesses:**

Pros:
- The paper is well organized and easy to follow
- Experiments studies to validate the proposed idea and other follow-up analysis

Cons:
Evaluation:
The proposed sample scoring scheme appears to be effective. However, I'm not entirely sure if the evaluation criteria is correct. For starters, the proposed sampler does uniform sampling for E epochs. Given this, its hard to verify how effective the  importance weighting scheme is.

The method is clearly sensitive to both E and \gamma. Experiments show that one needs to tune both these parameters for each model and each dataset. Whereas some of the baselines and other methods that are not compared don't need any hyper parameter tuning at all. Therefore claims about the proposed not needing additional training iterations aren't entirely fair in my opinion.

Class imbalance: I don't agree with the arguments in the section about class imbalance. The study there is based on the authors' proposed metric, which has an intuitive notion of "importance". Without rigorous theoretical grounding, the claims about class imbalance are not sound. These are however interesting observations and can benefit from thorough analysis.

**Summary Of The Paper:**

The authors propose a new importance sampling technique for SGD-based optimizers for training deep neural network models. Empirical studies are carried out on image classification tasks using standard benchmark datasets.

**Summary Of The Review:**

The authors propose an interesting sampling and training scheme. The experiments in the paper seems to suggest that the the weighting scheme is helping the SGD training. However, I believe the experimental setup and analysis don't clearly establish the benefits and issue of the proposed scheme.

---

### Decision · Program_Chairs · 2023-01-20

**Decision:**

Reject

**Justification For Why Not Higher Score:**

Concerns remained on lack of baselines and novelty

**Justification For Why Not Lower Score:**

N/A

**Metareview: Summary, Strengths And Weaknesses:**

The paper studies importance sampling for SGD training of neural networks. It follows the relatively standard approach of using first few epochs to compute importance scores, and later uses those fixed scores.

Unfortunately many concerns remained from the reviews both on the level of novelty and quality of the results. A main concern was that many existing baselines for importance sampling were not included in the experiments.

We hope the detailed feedback helps to strengthen the paper for a future occasion.